# Latte: Latent Diffusion Transformer for Video Generation

**Xin Ma**[1†], **Yaohui Wang**[2‡], **Xinyuan Chen**[2], **Gengyu Jia**[3],
**Ziwei Liu**[4], **Yuan-Fang Li**[1], **Cunjian Chen**[1] **Yu Qiao**[2]
[1]Department of Data Science & AI, Faculty of Information Technology, Monash University [2]Shanghai AI Laboratory
[3]Nanjing University of Posts and Telecommunications [4]S-Lab, Nanyang Technological University

**Reviewed on OpenReview:** `https://openreview.net/forum?id=ntGPYNUF3t`

## Abstract

We propose **Latte**, a novel *Latent Diffusion Transformer* for video generation. Latte first extracts spatio-temporal tokens from input videos and then adopts a series of Transformer blocks to model video distribution in the latent space. In order to model a substantial number of tokens extracted from videos, four efficient variants are introduced from the perspective of decomposing the spatial and temporal dimensions of input videos. To improve the quality of generated videos, we determine the best practices of Latte through rigorous experimental analysis, including video clip patch embedding, model variants, timestep-class information injection, temporal positional embedding, and learning strategies. Our comprehensive evaluation demonstrates that Latte achieves state-of-the-art performance across four standard video generation datasets, *i.e.*, FaceForensics, SkyTimelapse, UCF101, and Taichi-HD. In addition, we extend Latte to the text-to-video generation (T2V) task, where Latte achieves results that are competitive with recent T2V models. We strongly believe that Latte provides valuable insights for future research on incorporating Transformers into diffusion models for video generation. The project page is available at `https://maxin-cn.github.io/latte_project/`.

## 1 Introduction

Diffusion models Ho et al. (2020); Song et al. (2021b;a) are powerful deep generative models for many tasks in content creation, including image-to-image generation Meng et al. (2022); Zhao et al. (2022); Saharia et al. (2022a); Parmar et al. (2023), text-to-image generation Zhou et al. (2023); Rombach et al. (2022); Zhou et al. (2022); Ruiz et al. (2023); Zhang et al. (2023), and 3D object generation Wang et al. (2023); Chen et al. (2023); Zhou et al. (2021); Shue et al. (2023). Compared to these successful applications in images, generating high-quality videos still faces significant challenges, which can be primarily attributed to the intricate and high-dimensional nature of videos that encompass complex spatio-temporal information within high-resolution frames.

The significant role backbone models play in the success of diffusion models has also been investigated Nichol & Dhariwal (2021); Peebles & Xie (2023); Bao et al. (2023). The U-Net Ronneberger et al. (2015), which relies on Convolutional Neural Networks (CNNs), has held a prominent position in image and video generation works Ho et al. (2022); Dhariwal & Nichol (2021). In comparison, DiT Peebles & Xie (2023) and U-ViT Bao et al. (2023) adopt the architecture of ViT Dosovitskiy et al. (2021) in diffusion models for image generation and achieve great performance. Moreover, DiT has demonstrated that the inductive bias of U-Net is not crucial for the performance of latent diffusion models. On the other hand, attention-based architectures Vaswani et al. (2017) present an intuitive option for capturing long-range contextual relationships in videos. Therefore, a very natural question arises: *Can Transformer-based latent diffusion models enhance the generation of realistic videos?*

---

‡ Corresponding author. † Intern at Shanghai AI Lab.

Figure 1: **Sample videos with a resolution of 512 × 512**. Latte can generate photorealistic videos with temporal coherent content. Please click the image to play the video clip via Acrobat Reader.

In this paper, we propose **Latte**, the novel latent diffusion Transformers for video generation, which adopts a video Transformer as the backbone. Latte employs a pre-trained variational autoencoder to encode input videos into features in the latent space, where tokens are extracted from encoded features. Thus, a series of transformer blocks is applied to encode these tokens. Considering the inherent disparities between spatial and temporal information and a large number of tokens extracted from input videos, as shown in Fig. 2, we design four efficient Transformer-based model variants from the perspective of disentangling the spatial and temporal dimensions of input videos. We also give more insights about the impact of the degree of decoupling between the temporal and spatial modules within the different models, which are lacking in previous works Lu et al. (2024); Gupta et al. (2024).

Many best practices for convolutional models have been proposed, including text representation for question classification Pota et al. (2020), and network architecture design for image classification He et al. (2016), etc. Nevertheless, Transformer-based latent diffusion models for video generation might demonstrate different characteristics, necessitating the identification of optimal design choices for this architecture. Therefore, we conduct a comprehensive ablation analysis, encompassing *video clip patch embedding*, *model variants*, *timestep-class information injection*, *temporal positional embedding*, and *learning strategies*. Our analysis identifies best practices that enable Latte to generate photorealistic videos with temporal coherent content (see Fig. 1) and achieve state-of-the-art performance across four standard video generation benchmarks, including FaceForensics Rössler et al. (2018), SkyTimelapse Xiong et al. (2018), UCF101 Soomro et al. (2012) and Taichi-HD Siarohin et al. (2019) in terms of Fréchet Video Distance (FVD) Unterthiner et al. (2018), Fréchet Inception Distance (FID) Parmar et al. (2021), and Inception Score (IS). In addition, we extend Latte to text-to-video generation tasks, which also achieve competitive results compared to current T2V models.

Our main contributions can be summarized as follows:

- We present Latte, a novel latent diffusion Transformer, which adopts a video Transformer as the backbone. In addition, four model variants are introduced to efficiently capture spatio-temporal distribution in videos. The design insights of four model variants are also analyzed in Sec. 4.2.

- To improve the quality of generated videos, we comprehensively explore design alternatives, including video clip patch embedding, model variants, timestep-class information injection, temporal positional

embedding, and learning strategies, to determine the best practices of Transformer-based diffusion models for video generation.

- Experimental results on four standard video generation benchmarks show that Latte can generate photorealistic videos with temporal coherent content outperforming state-of-the-art methods. Moreover, Latte shows competitive results when applied to the text-to-video generation task.

## 2 Related Work

**Video generation** aims to produce realistic videos that exhibit a high-quality visual appearance and consistent motion simultaneously. Previous research in this field can be categorized into three main categories. Firstly, several studies have sought to extend the capabilities of powerful GAN-based image generators to create videos Vondrick et al. (2016); Saito et al. (2017); Wang et al. (2020b;a); Kahembwe & Ramamoorthy (2020). However, these methods often encounter challenges related to mode collapse, limiting their effectiveness. Secondly, some methods propose learning the data distribution using autoregressive models Ge et al. (2022); Rakhimov et al. (2021); Weissenborn et al. (2020); Yan et al. (2021); Kondratyuk et al. (2024). While these approaches generally offer good video quality and exhibit more stable convergence, they come with the drawback of requiring significant computational resources. Finally, recent advances in video generation have focused on building systems based on diffusion models Ho et al. (2020); Harvey et al. (2022); Ho et al. (2022); Singer et al. (2023); Mei & Patel (2023); Blattmann et al. (2023b); Wang et al. (2024a); Chen et al. (2024c); Wang et al. (2024b); Gupta et al. (2024); Ma et al. (2024a), resulting in promising outcomes. However, Transformer-based diffusion models have not been explored well. A similar idea has been explored in recent concurrent work VDT Lu et al. (2024), GenTron Chen et al. (2024b), Open-Sora Plan Lin et al. (2024), HunyuanVideo Kong et al. (2024), Pyramidal Flow Jin et al. (2025), Cosmos Agarwal et al. (2025) and so on. VDT, GenTron, and W.A.L.T use an architecture akin to our Variant 3. The primary difference from the previous works is that we conduct a systematic analysis of different Transformer backbones and the relative best practices discussed in Sec. 3.2 and Sec. 3.3 on video generation. Please see Sec. A.3 for a more detailed discussion of the differences.

**Transformers** have become the mainstream model architecture and got remarkable success in many fields, such as image inpainting Ma et al. (2022; 2021; 2023), image super-resolution Luo et al. (2022); Huang et al. (2017), image cropping Jia et al. (2022), forgery detection Jia et al. (2021), face recognition Luo et al. (2021a;b), natural language processing Devlin et al. (2019); Ma et al. (2024b). Transformers initially emerged within the language domain Vaswani et al. (2017); Kaplan et al. (2020), where they quickly established a reputation for their outstanding capabilities. Over time, these models have been adeptly adapted for the task of predicting images, performing this function autoregressively within both image spaces and discrete codebooks Chen et al. (2020); Parmar et al. (2018). In the latest developments, Transformers have been integrated into diffusion models, expanding their purview to the generation of non-spatial data and images. This includes tasks like text encoding and decoding Rombach et al. (2022); Saharia et al. (2022b), generating CLIP embedding Ramesh et al. (2022), and realistic image generation Bao et al. (2023); Peebles & Xie (2023).

## 3 Methodology

We begin with a brief introduction of latent diffusion models in Sec. 3.1. Following that, we present the model variants of Latte in Sec. 3.2. Finally, the architectural design best practices are discussed in Sec. 3.3.

### 3.1 Preliminary of Latent Diffusion Models

**Latent diffusion models (LDMs)** Rombach et al. (2022). LDMs are efficient diffusion models Ho et al. (2020); Song et al. (2021b) by conducting the diffusion process in the latent space instead of the pixel space. LDMs first employ an encoder $\mathcal{E}$ from a pre-trained variational autoencoder to compress the input data sample $x \in p_{\text{data}}(x)$ into a lower-dimensional latent code $z = \mathcal{E}(x)$. Subsequently, it learns the data distribution through two key processes: diffusion and denoising.

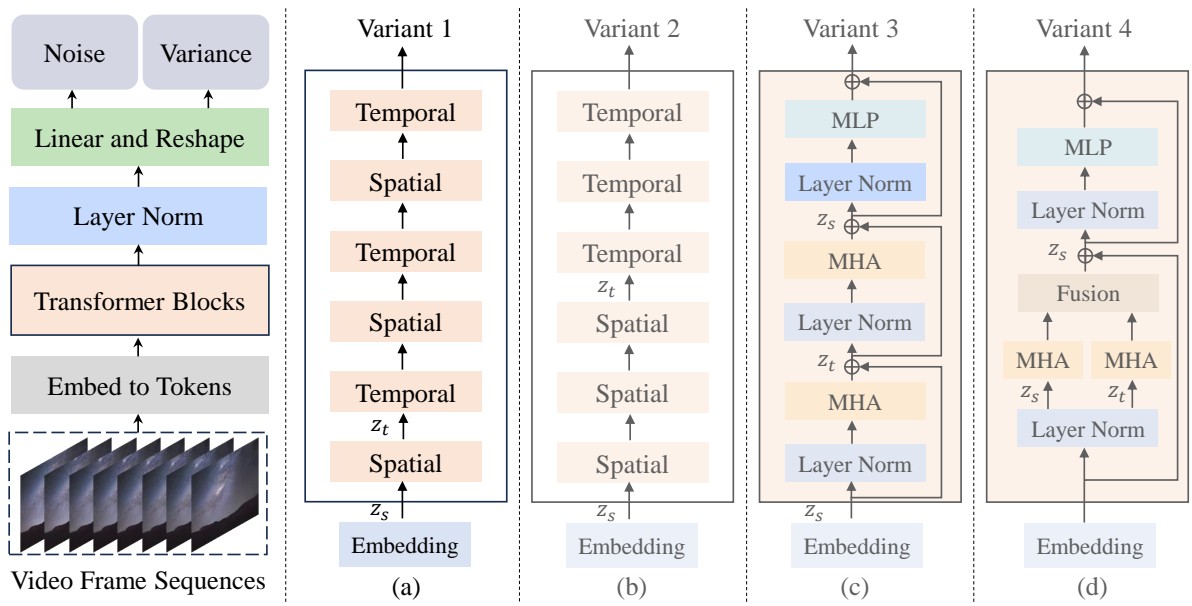

Figure 2: **The Latte architecture for video generation**. Four model variants of Latte are proposed to capture spatio-temporal information in videos efficiently. Each block depicted in light orange represents a Transformer block. The standard Transformer blocks (described in Fig. 10 of the Appendix) are employed in (a) and (b). Meanwhile, (c) and (d) employ our custom Transformer block variants. For the sake of simplicity, encoding and decoding processes for VAE are not shown in the diagram.

The diffusion process gradually introduces Gaussian noise into the latent code $z$, generating a perturbed sample $z_t = \sqrt{\overline{\alpha}_t}z + \sqrt{1 - \overline{\alpha}_t}\epsilon$, where $\epsilon \sim \mathcal{N}(0, 1)$, following a Markov chain spanning $T$ stages. In this context, $\overline{\alpha}_t$ serves as a noise scheduler, with $t$ representing the diffusion timestep.

The denoising process is trained to understand the inverse diffusion process to predict a less noisy $z_{t-1}$: $p_\theta(z_{t-1}|z_t) = \mathcal{N}(\mu_\theta(z_t), \Sigma_\theta(z_t))$ with the variational lower bound of log-likelihood reducing to $\mathcal{L}_\theta = -\log p(z_0|z_1) + \sum_t D_{KL}((q(z_{t-1}|z_t, z_0)||p_\theta(z_{t-1}|z_t))$. Here, $\mu_\theta$ is implemented using a denoising model $\epsilon_\theta$ and is trained with the *simple* objective,

$$\mathcal{L}_{simple} = \mathbb{E}_{\mathbf{z} \sim p(z), \ \epsilon \sim \mathcal{N}(0,1), \ t} \left[ \|\epsilon - \epsilon_\theta(\mathbf{z}_t, t)\|_2^2 \right]. \tag{1}$$

In accordance with Nichol & Dhariwal (2021), to train diffusion models with a learned reverse process covariance $\Sigma_\theta$, it is necessary to optimize the full $D_{KL}$ term and thus train with the full loss function, denoted as $\mathcal{L}_{vlb}$. Additionally, $\Sigma_\theta$ is implemented using $\epsilon_\theta$.

### 3.2 The Latte architecture and its model variants

In this work, we propose Latte, a Transformer-based architecture, shown in Fig. 2 (left), for video generation. Latte extends LDMs in the following ways. (1) The encoder $\mathcal{E}$ is used to compress each video frame into latent space. (2) The diffusion process operates in the latent space of videos to model the latent spatial and temporal information. $\epsilon_\theta$ is implemented with a Transformer. We train all our models by employing both $\mathcal{L}_{simple}$ and $\mathcal{L}_{vlb}$. We propose four model variants of Latte to efficiently capture spatio-temporal information in videos. The variants are depicted in Fig. 2 (right).

**Variant 1.** As depicted in Fig. 2 (a), the Transformer backbone of this variant comprises two distinct types of Transformer blocks: spatial Transformer blocks and temporal Transformer blocks. The former focuses on capturing spatial information exclusively among tokens sharing the same temporal index, while the latter captures temporal information across temporal dimensions in an "interleaved fusion" manner.

Given a video clip in the latent space $\boldsymbol{V_L} \in \mathbb{R}^{F \times H \times W \times C}$, we first translate $\boldsymbol{V_L}$ into a sequence of tokens, denoted as $\hat{\boldsymbol{z}} \in \mathbb{R}^{n_f \times n_h \times n_w \times d}$. Here, $F$, $H$, $W$, and $C$ represent the number of video frames, the height, width, and channel of video frames in the latent space, respectively. The total number of tokens within a video clip in the latent space is $n_f \times n_h \times n_w$, and $d$ represents the dimension of each token, respectively. Spatio-temporal positional embedding $\boldsymbol{p}$ is incorporated into $\hat{\boldsymbol{z}}$. Finally, we get the $\boldsymbol{z} = \hat{\boldsymbol{z}} + \boldsymbol{p}$ as the input for the Transformer backbone.

We reshape $\boldsymbol{z}$ into $\boldsymbol{z_s} \in \mathbb{R}^{n_f \times s \times d}$ as the input of the spatial Transformer block to capture spatial information. Here, $s = n_h \times n_w$ denotes the token count of each temporal index. Subsequently, $\boldsymbol{z_s}$ containing spatial information is reshaped into $\boldsymbol{z_t} \in \mathbb{R}^{s \times n_f \times d}$ to serve as the input for the temporal Transformer block, which is used for capturing temporal information.

**Variant 2.** In contrast to the temporal "interleaved fusion" design in Variant 1, this variant utilizes the "late fusion" approach to combine spatio-temporal information Neimark et al. (2021); Simonyan & Zisserman (2014). As depicted in Fig. 2 (b), this variant consists of an equal number of Transformer blocks as in Variant 1. Similar to Variant 1, the input shapes for the spatial Transformer block and temporal Transformer block are $\boldsymbol{z_s} \in \mathbb{R}^{n_f \times s \times d}$ and $\boldsymbol{z_t} \in \mathbb{R}^{s \times n_f \times d}$ respectively.

**Variant 3.** Variant 1 and Variant 2 primarily focus on the factorization of the Transformer blocks. Variant 3 focuses on decomposing the multi-head attention in the Transformer block. Illustrated in Fig. 2 (c), this variant initially computes self-attention only on the spatial dimension, followed by the temporal dimension. As a result, each Transformer block captures both spatial and temporal information. Similar to Variant 1 and Variant 2, the inputs for spatial multi-head self-attention and temporal multi-head self-attention are $\boldsymbol{z_s} \in \mathbb{R}^{n_f \times s \times d}$ and $\boldsymbol{z_t} \in \mathbb{R}^{s \times n_f \times d}$, respectively.

**Variant 4.** We decompose the multi-head attention (MHA) into two components in this variant, with each component utilizing half of the attention heads as shown in Fig. 2 (d). We use different components to handle tokens separately in spatial and temporal dimensions. The input shapes for these different components are $\boldsymbol{z_s} \in \mathbb{R}^{n_f \times s \times d}$ and $\boldsymbol{z_t} \in \mathbb{R}^{s \times n_f \times d}$ respectively. Once two different attention operations are calculated, we reshape $\boldsymbol{z_t} \in \mathbb{R}^{s \times n_f \times d}$ into $\boldsymbol{z_t'} \in \mathbb{R}^{n_f \times s \times d}$. Then $\boldsymbol{z_t'}$ is added to $\boldsymbol{z_s}$, which is used as the input for the next module in the Transformer block.

After the Transformer backbone, a critical procedure involves decoding the video token sequence to derive both predicted noise and predicted covariance. The shape of the two outputs is the same as that of the input $\boldsymbol{V_L} \in \mathbb{R}^{F \times H \times W \times C}$. Following previous work Peebles & Xie (2023); Bao et al. (2023), we accomplish this by employing a standard linear decoder as well as a reshaping operation. We also provide more design signing about the four different model variants and show why variant 1 achieves the best performance in Sec. 4.2.

## 3.3 The Architectural Design Best Practices of Latte

We perform a comprehensive empirical analysis of crucial components in Latte, aiming to discover the best practices for integrating the Transformer as the backbone within latent diffusion models for video generation.

### 3.3.1 Latent video clip patch embedding

To embed a video clip, we explore two methods as follows to analyze the necessity of integrating temporal information in tokens, *i.e.* 1) uniform frame patch embedding and 2) compression frame patch embedding.

**Uniform frame patch embedding.** As illustrated in Fig. 3 (a), we apply the patch embedding technique outlined in ViT Dosovitskiy et al. (2021) to each video frame individually. Specifically, $n_f$, $n_h$, and $n_w$ are equivalent to $F$, $\frac{H}{h}$, and $\frac{W}{w}$ when non-overlapping image patches are extracted from every video frame. Here, $h$ and $w$ denote the height and width of the image patch, respectively.

**Compression frame patch embedding.** The second approach is to model the temporal information in a latent video clip by extending the ViT patch embedding to the temporal dimension, as shown in Fig. 3 (b). We extract tubes along the temporal dimension with a stride of $s$ and then map them to tokens. Here, $n_f$ is equivalent to $\frac{F}{s}$ in contrast to non-overlapping uniform frame patch embedding. Compared to the former, this method inherently incorporates spatio-temporal information during the patch embedding stage.

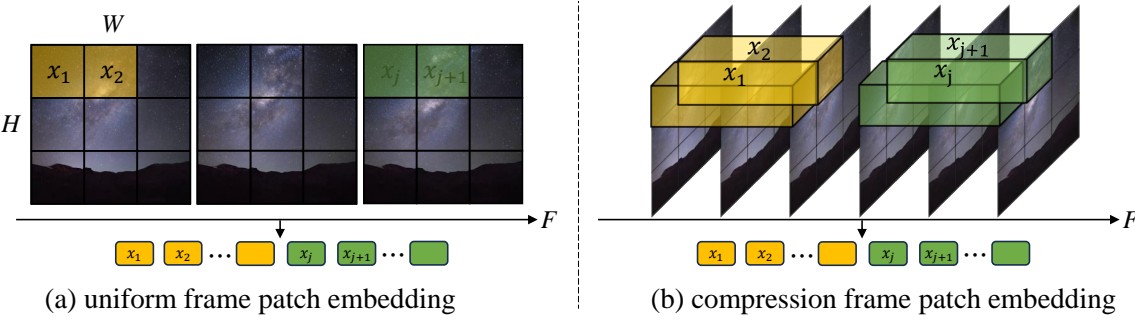

Figure 3: **The video clip patch embedding**. (a) We sample $F$ frames and embed each individual video frame into tokens using the method described in ViT. (b) We consider capturing temporal information and then extending the ViT patch embedding method from 2D to 3D and subsequently extracting tubes along the temporal dimension. For ease of understanding, we use the original video clip here to demonstrate the patch embedding method. The patch embedding in the latent space of videos follows the same processing approach.

Note that in the context of using the compression frame patch embedding method, an additional step entails integrating a 3D transposed convolution for temporal upsampling of the output latent videos, following the standard linear decoder and reshaping operation.

### 3.3.2 Timestep-class information injection

From simple and direct integration to complex and nuanced integration perspective, we explore two methods for integrating timestep or class information $c$ into our model. The first approach involves treating it as tokens, and we refer to this approach as *all tokens*. The second method is akin to adaptive layer normalization (AdaLN) following DiT Perez et al. (2018); Peebles & Xie (2023). We employ linear regression to compute $\gamma_c$ and $\beta_c$ based on the input $c$, resulting in the equation $AdaLN(h,c) = \gamma_c\text{LayerNorm}(h) + \beta_c$, where $h$ represents the hidden embeddings within the Transformer blocks. Furthermore, we also perform regression on $\alpha_c$, which is applied directly before any residual connections (RCs) within the Transformer block, resulting in $RCs(h,c) = \alpha_c h + AdaLN(h,c)$. We refer to this as scalable adaptive layer normalization ($S\text{-}AdaLN$). The architecture of $S\text{-}AdaLN$ is shown in Fig. 10 of the Appendix.

### 3.3.3 Temporal positional embedding

Temporal positional embedding enables a model to comprehend the temporal signal. We explore two methods as follows for injecting temporal positional embedding into the model: 1) the absolute positional encoding method incorporates sine and cosine functions with varying frequencies Vaswani et al. (2017) to enable the model to recognize the precise position of each frame within the video sequence; 2) the relative positional encoding method employs rotary positional embedding (RoPE) Su et al. (2024) to enable the model to grasp the temporal relationships between successive frames.

### 3.3.4 Enhancing video generation with learning strategies

Our goal is to ensure that the generated videos exhibit the best visual quality while preserving temporal consistency. We explore whether incorporating two additional learning strategies, i.e., learning with pre-trained models and learning with image-video joint training, can enhance the quality of the generated videos.

**Learning with pre-trained models.** The pre-trained image generation models have learned what the world looks like. Thus, there are many video generation works that ground their models on pre-trained image generation models to learn how the world moves Wang et al. (2024a); Blattmann et al. (2023a). However, these works mainly build on U-Net within latent diffusion models. The necessity of Transformer-based latent diffusion models is worth exploring.

We initialize Latte from a pre-trained DiT model on ImageNet Peebles & Xie (2023); Deng et al. (2009). Directly initializing from the pre-trained DiT model will encounter the problem of missing or incompatible parameters. To address these, we implement the following strategies. In pre-trained DiT, a positional embedding $\boldsymbol{p} \in \mathbb{R}^{n_h \times n_w \times d}$ is applied to each token. However, in our video generation model, we have a token count that is $n_f$ times greater than that of the pre-trained DiT model. We thus temporally replicate the positional embedding $n_f$ times from $\boldsymbol{p} \in \mathbb{R}^{n_h \times n_w \times d}$ to $\boldsymbol{p} \in \mathbb{R}^{n_f \times n_h \times n_w \times d}$. Furthermore, the pre-trained DiT includes a label embedding layer, and the number of categories is 1000. Nevertheless, the used video dataset either lacks label information or encompasses a significantly smaller number of categories in comparison to ImageNet. Since we target both unconditional and class-conditional video generation, the original label embedding layer in DiT is inappropriate for our tasks, we opt to directly discard the label embedding in DiT and apply zero-initialization.

**Learning with image-video joint training.** The prior work on the CNN-based video diffusion model proposes a joint image-video training strategy that greatly improves the quality of the generated videos Ho et al. (2022). We explore whether this training strategy can also improve the performance of the Transformer-based video diffusion model. To implement simultaneous training for video and image generation, We append randomly selected video frames from the same dataset to the end of the chosen videos, and each frame is independently sampled. In order to ensure our model can generate continuous videos, we propose two image-video joint training strategies: 1) tokens related to video content are used in the temporal module for modeling temporal information, while video frames tokens are excluded; 2) all tokens are used to optimize the temporal module with a new masking strategy. The first mask training strategy is that tokens related to video content are used in the temporal module to establish temporal relationships between video frames, referred to as *Only Video*. However, the performance is not optimal since images are only used to optimize the spatial modules. We thus propose a new training strategy that can make images also optimize the temporal modules while keeping our model to generate consistent videos, referred to as *Temporal Mask*. Suppose we have $f$ frames in a video clip and $i$ additional images at the end of the video, thus we have a mask matrix $M \in \mathbb{R}^{(f+i) \times (f+i)}$. The visualization of our temporal mask $M$ can be seen in Fig. 4. We evaluate the zero-shot capability on UCF101 using two different mask strategies. The FVD and FID scores for *Only Video* are 705.72 and 75.98, respectively, while for *Temporal Mask*, they are 589.86 and 63.36. These results demonstrate the superiority of *Temporal Mask*.

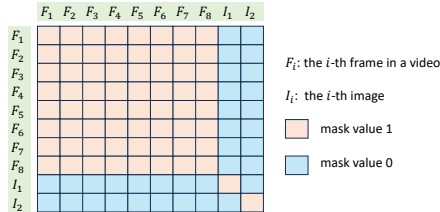

Figure 4: Temporal mask strategy.

## 4 Experiments

This section initially outlines the experimental setup, encompassing datasets, evaluation metrics, baselines, Latte configurations, and implementation details. Subsequently, we present ablation experiments for the best practice choices and model size of Latte. Finally, we compare experimental results with state-of-the-art and present text-to-video generation results.

### 4.1 Experimental setup

**Datasets.** We primarily conduct comprehensive experiments on four public datasets: FaceForensics Rössler et al. (2018), SkyTimelapse Xiong et al. (2018), UCF101 Soomro et al. (2012), and Taichi-HD Siarohin et al. (2019). Following the experimental setup in Skorokhodov et al. (2022), except for UCF101, we use the training split for all datasets if they are available. For UCF101, we use both training and testing splits. We extract 16-frame video clips from these datasets using a specific sampling interval, with each frame resized to 256×256 resolution for training.

**Evaluation metrics.** In the assessment of quantitative comparisons, we employ three evaluation metrics: Fréchet Video Distance (FVD) Unterthiner et al. (2018), Fréchet Inception Distance (FID) Parmar et al. (2021), and Inception Score (IS) Saito et al. (2017). Our primary focus rests on FVD, as its image-based counterpart FID aligns more closely with human subjective judgment. Adhering to the evaluation guidelines

|  | Variant 1 | Variant 2 | Variant 3 | Variant 4 |
|---|---|---|---|---|
| Params (M) | 673.68 | 673.68 | 676.33 | 676.44 |
| FLOPs (G) | 5572.69 | 5572.69 | 6153.15 | 1545.15 |

Table 1: The number of parameters and FLOPs for different model variants.

| Model | $N$ | $D$ | $H$ | Param |
|---|---|---|---|---|
| Latte-S | 12 | 384 | 6 | 32.48M |
| Latte-B | 12 | 768 | 12 | 129.54M |
| Latte-L | 24 | 1024 | 16 | 456.81M |
| Latte-XL | 28 | 1152 | 16 | 673.68M |

Table 2: **Details of Latte models.** We follow ViT and DiT configurations.

| Method | FaceForensics | SkyTimelapse | UCF101 | Taichi-HD |
|---|---|---|---|---|
| MoCoGAN Tulyakov et al. (2018) | 124.7 | 206.6 | 2886.9 | - |
| VideoGPT Yan et al. (2021) | 185.9 | 222.7 | 2880.6 | - |
| MoCoGAN-HD Tian et al. (2021) | 111.8 | 164.1 | 1729.6 | 128.1 |
| DIGAN Yu et al. (2022) | 62.5 | 83.11 | 1630.2 | 156.7 |
| StyleGAN-V Skorokhodov et al. (2022) | 47.41 | 79.52 | 1431.0 | - |
| PVDM Yu et al. (2023) | 355.92 | 55.41 | 343.6 | 540.2 |
| MoStGAN-V Shen et al. (2023) | 39.70 | 65.30 | 1380.3 | - |
| LVDM He et al. (2023) | - | 95.20 | 372.0 | 99.0 |
| VDT Lu et al. (2024) | - | - | 225.7 | - |
| W.A.L.T Gupta et al. (2024) | - | - | 258.1 | - |
| Latte (ours) | 34.00 | 59.82 | 477.97 | 159.60 |
| Latte+IMG (ours) | **27.08** | **42.67** | **202.23** | **97.09** |

Table 3: **FVD values of video generation models on different datasets**. FVD values for other baseline models are reported and sourced from the reference StyleGAN-V or the original paper. Here, "IMG" means video-image joint training.

| Methods | Subject Consistency | Background Consistency | Temporal Flickering | Motion Smoothness | Dynamic Degree | Aesthetic Quality | Imaging Quality | Object Class |
|---|---|---|---|---|---|---|---|---|
| ModelScope Luo et al. (2023) | 89.87% | 95.29% | 98.28% | 95.79% | 66.39% | 52.06% | 58.57% | 82.25% |
| VideoCrafter He et al. (2023) | 86.24% | 92.88% | 97.60% | 91.79% | 89.72% | 44.41% | 57.22% | **87.34%** |
| CogVideo Hong et al. (2022) | 92.19% | 95.42% | 97.64% | 96.47% | 42.22% | 38.18% | 41.03% | 73.40% |
| HiGen Qing et al. (2024) | 90.07% | 93.99% | 93.24% | 96.69% | **99.17%** | 57.30% | 63.92% | 86.06% |
| OpenSoraPlan Lin et al. (2024) | **97.27%** | **96.24%** | **99.12%** | **99.17%** | 35.28% | 59.10% | **65.73%** | 61.53% |
| Latte | 88.88% | 95.40% | 98.89% | 94.63% | 68.89% | **61.59%** | 61.92% | 86.53% |

| Methods | Multiple Object | Human Action | Color | Spatial Relationship | Scene | Appearance Style | Temporal Style | Overall Consistency |
|---|---|---|---|---|---|---|---|---|
| ModelScope Luo et al. (2023) | **38.98%** | 92.40% | 81.72% | 33.68% | 39.26% | 23.39% | 25.37% | 25.67% |
| VideoCrafter He et al. (2023) | 25.93% | **93.00%** | 78.84% | 36.74% | 43.36% | 21.57% | **25.42%** | 25.21% |
| CogVideo Hong et al. (2022) | 18.11% | 78.20% | 79.57% | 18.24% | 28.24% | 22.01% | 7.80% | 7.70% |
| HiGen Qing et al. (2024) | 22.39% | 86.20% | 86.22% | 22.43% | **44.88%** | **24.54%** | 25.14% | 27.14% |
| OpenSoraPlan Lin et al. (2024) | 24.91% | 58.20% | **90.90%** | **49.64%** | 17.28% | 20.04% | 19.14% | 20.39% |
| Latte | 34.53% | 90.00% | 85.31% | 41.53% | 36.26% | 23.74% | 24.76% | **27.33%** |

Table 4: **Vbench evaluation results per dimension for different methods.** A higher score demonstrates better model performance for a certain dimension. We use **bold** and underline to mark the best and second model performances, respectively.

introduced by StyleGAN-V, we compute the FVD scores by analyzing 2,048 video clips, each comprising 16 frames. We only employ IS for assessing the generation quality on UCF101, as it leverages the UCF101-fine-tuned C3D model Saito et al. (2017).

**Baselines.** We compare with recent methods to quantitatively evaluate the outcomes, including MoCoGAN Tulyakov et al. (2018), VideoGPT Yan et al. (2021), MoCoGAN-HD Tian et al. (2021), DIGAN Yu et al. (2022), StyleGAN-V Skorokhodov et al. (2022), PVDM Yu et al. (2023), MoStGAN-V Shen et al. (2023), LVDM He et al. (2023), VDT Lu et al. (2024) and W.A.L.T Gupta et al. (2024). Furthermore, we conduct an extra comparison of IS between our proposed method and previous approaches on the UCF101 dataset.

**Latte configurations.** A series of $N$ Transformer blocks are used to construct our Latte model, and the hidden dimension of each Transformer block is $D$ with $N$ multi-head attention. Following ViT, we identify four configurations of Latte with different numbers of parameters as shown in Tab. 2.

**Implementation details.** We use the AdamW optimizer with a constant learning rate $1 \times 10^{-4}$ to train all models. Horizontal flipping is the only employed data augmentation. Following common practices within generative modeling works Peebles & Xie (2023); Bao et al. (2023), an exponential moving average (EMA) of Latte weights is upheld throughout training, employing a decay rate of 0.9999. All the reported results are directly obtained from the EMA model. We borrow the pre-trained variational autoencoder from Stable Diffusion 1.4. All experiments are conducted on 8 NVIDIA A100 (80G) GPUs.

## 4.2 Ablation study

In this section, we conduct experiments on the FaceForensics dataset to examine the effects of different designs described in Sec. 3.3, model variants described in Sec. 3.2, video sampling interval, and model size on model performance.

**Video clip patch embedding.** We examine the impact of two video clip patch embedding methods detailed in Sec 3.3.1. In Fig. 5a, the performance of the compression frame patch embedding method notably falls behind that of the uniform frame patch embedding method. This finding contradicts the results obtained by the video understanding method ViViT Arnab et al. (2021). We speculate that using the compression frame patch embedding method results in the loss of spatio-temporal signal, which makes it difficult for the Transformer backbone to learn the distribution of videos.

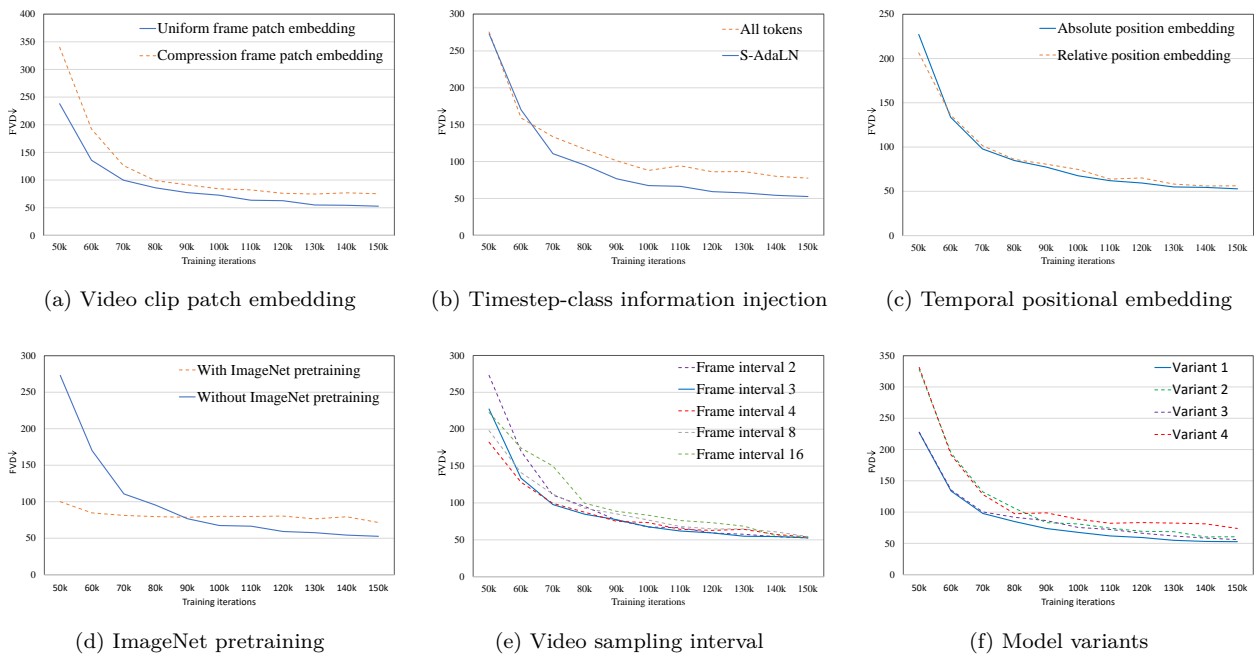

(a) Video clip patch embedding  (b) Timestep-class information injection  (c) Temporal positional embedding

(d) ImageNet pretraining  (e) Video sampling interval  (f) Model variants

Figure 5: **Ablation of design choices**. We design several ablation studies to explore best practices in Transformer-based video diffusion models in terms of FVD on FaceForensics. Please zoom in for a better view.

**Timestep-class information injection.** As depicted in Fig. 5b, the performance of *S-AdaLN* is significantly better than that of *all tokens*. We believe this discrepancy may stem from the fact that *all tokens* only introduces timesteps or label information to the input layer of the model, which could face challenges in propagating effectively throughout the model. In contrast, *S-AdaLN* encodes timestep or label informa-

tion into the model in a more adaptive manner for each Transformer block. This information transmission approach appears more efficient, likely contributing to superior performance and faster model convergence.

**Temporal positional embedding.** Fig. 5c illustrates the impact of two different temporal position embedding methods on the performance of the model. Employing the absolute position embedding approach tends to yield slightly better results than the alternative method. Since our model does not require dynamic temporal-spatial adaptability for generating variable-duration and multi-resolution videos (i.e., a key strength of RoPE), we ultimately choose absolute positional embedding as the temporal positional embedding method.

**Enhancing video generation with learning strategies.** As illustrated in Fig. 5d, we observe that the initial stages of training benefit greatly from the model pre-training on ImageNet, enabling rapid achievement of high-quality performance on the video dataset. However, as the number of iterations increases, the performance of the model initialized with a pre-trained model tends to stabilize around a certain level, which is far worse than that of the model initialized with random.

This phenomenon can be explained by two factors: 1) the pre-trained model on ImageNet provides a good representation, which may help the model converge quickly at an early stage; 2) there is a significant difference in data distribution between ImageNet and FaceForensics, which makes it difficult for the model to adapt the knowledge learned on ImageNet to FaceForensics. It is important to emphasize that the second point is particularly relevant for small-scale datasets (e.g., UCF101), where the domain discrepancy may outweigh the benefits of knowledge transfer. In contrast, large-scale datasets inherently exhibit greater diversity, which helps mitigate domain gaps compared to specialized small datasets. As a result, the first factor may play a more significant role in general tasks, such as text-to-video generation.

As demonstrated in Tab. 3 and Tab. 5, we find that image-video joint training ("Latte+IMG") leads to a significant improvement of FID and FVD. Concatenating additional randomly sampled frames with videos along the temporal axis enables the model to accommodate more examples within each batch, which can increase the diversity of trained models. **Video sampling interval.** We explore various sampling rates to construct a 16-frame clip from each training video. As illustrated in Fig. 5e, during training, there is a significant performance gap among models using different sampling rates in the early stages. However, as the number of training iterations increases, the performance gradually becomes consistent, which indicates that different sampling rates have little effect on model performance. We choose a video sampling interval of 3 to ensure a reasonable level of continuity in the generated videos to conduct the experiments of comparison to state-of-the-art.

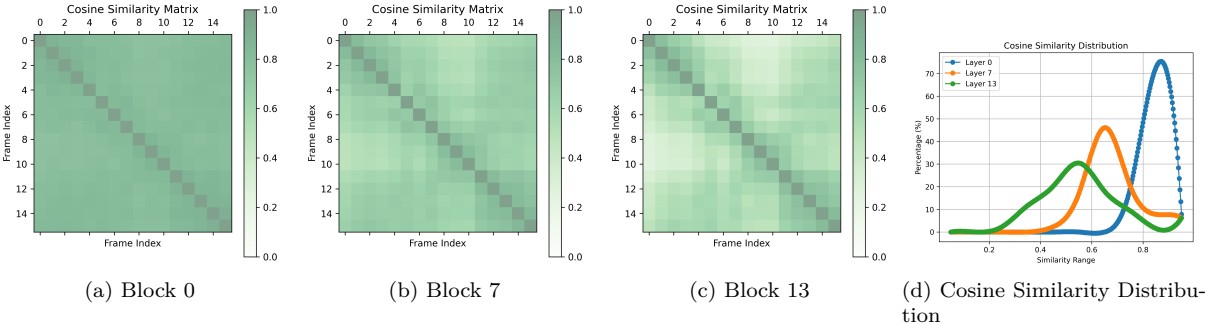

(a) Block 0          (b) Block 7          (c) Block 13          (d) Cosine Similarity Distribution

Figure 6: Frame cosine similarity and its distribution.

**Model variants.** We evaluate the model variants of Latte as detailed in Sec. 3.2. We strive to equate the parameter counts across all different models to ensure a fair comparison. We commence training all the models from scratch. As shown in Fig. 5f, Variant 1 performs the best with increasing iterations. Notably, Variant 4 exhibits roughly a quarter of the floating-point operations (FLOPs) compared to the other three model variants, as detailed in Tab. 1. Therefore, it is unsurprising that Variant 4 performs the least favorably among the four variants.

In Variant 2, half of the Transformer blocks are initially employed for spatial modeling, followed by the remaining half for temporal modeling. Such division may lead to the loss of spatial modeling capabilities during subsequent temporal modeling, ultimately impacting performance. Hence, we think employing a complete Transformer block (including multi-head attention, layer norm, and multi-linear projection) might be more effective in modeling temporal information compared to only using multi-head attention (Variant 3).

**Model size.** We train four Latte models of different sizes according to Tab. 2 (XL, L, B, and S) on the FaceForensics dataset. Fig. 7 clearly illustrates the progression of corresponding FVDs as the number of training iterations increases. It can be clearly observed that increasing the model size generally correlates with a notable performance improvement, which has also been pointed out in image generation work Peebles & Xie (2023).

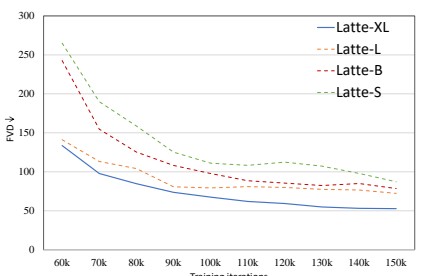

Figure 7: **The model performance of different Latte model sizes**. In general, increasing the size of the model can significantly improve its performance.

**Design insights for four model variants.** The design of the four variants stems from a key question: to what extent should the temporal and spatial modules be decoupled in a video generation model? While similar model structures have been employed in some previous works Gupta et al. (2024); Lu et al. (2024), they have not analyzed how these different model structures affect the video generation process. We have supplemented this with detailed experiments and analyses to help understand the impact of the degree of decoupling between the temporal and spatial modules and to explain why Variant 1 achieves the best performance.

*Excessive consecutive spatial attention modules can impair temporal coherence.* As shown in Fig. 6, we conduct a detailed analysis of Variant 2, calculating the cosine similarity matrix between frame features at each block. The results demonstrate that adding spatial attention modules leads to a continuous decrease in the mean of inter-frame cosine similarity and an increase in variance. This uneven decrease in inter-frame cosine similarity ultimately results in numerous *inverted relationships* in the cosine similarity matrix, where, for instance, the cosine similarity between the first and second frames is lower than that between the first and fifth frames. These *inverted relationships* are strong evidence of the disruption of temporal coherence. We believe the root cause of this phenomenon lies in the coupled temporal-spatial information failing to achieve effective decoupling through the separation of spatiotemporal modules.

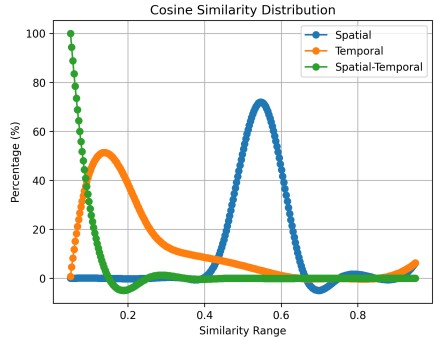

Figure 8: The distribution of frame cosine similarity.

*The features of the output frames from spatial attention do not match those from temporal attention.* The above analysis disproves the design of stacking massive spatial attention modules, leading us to explore increasing the coupling degree between spatial and temporal modules. To this end, we design Variant 4, which modifies the alternating structure of spatiotemporal attention to a parallel structure within the Transformer block. However, experiments show that this model structure performs poorly. Analysis reveals that the performance degradation is due to the direct fusion of mismatched frame features. Specifically, as shown in Fig. 8, we visualize the distributions of inter-frame cosine similarity for the outputs of temporal attention, spatial attention, and their mutual inter-frame cosine similarity. The results show significant differences in the distributions of inter-frame cosine similarity for each type, with extremely low cosine similarity between the two. Clearly, directly adding these two types of features severely disrupts temporal information.

The design of Variant 3 is derived from TimesFormer Bertasius et al. (2021). Unlike Variant 1, which also employs an alternating spatiotemporal attention structure, Variant 3 has a higher degree of coupling. The

spatial attention features are input into the temporal attention module, and only layer normalization is applied. Based on the above analysis, we believe this structure is not optimal. The simple normalization of spatial attention frame features is insufficient to make them suitable for the temporal attention layer. Experiments show that while Variant 3 performs better than Variants 2 and 4, it is weaker than Variant 1. This result further confirms the validity of our analysis.

### 4.3 Summary of best practices

Based on the ablation studies in Section 4.2, we identify the best practices for Transformer-based latent video diffusion models. These include model variant 1, uniform frame patch embedding, *S-AdaLN*, absolute position embedding, and image-video joint training. In the next section, we compare our proposed Latte, incorporating these best practices, with the current state-of-the-art.

### 4.4 Comparison to state-of-the-art

**Qualitative results.** Fig. 9 of Appendix illustrates the video synthesis results from Latte on UCF101, Taichi-HD, FaceForensics, and SkyTimelapse. Our method consistently delivers realistic, high-resolution video generation results (256x256 pixels) in all scenarios. This encompasses capturing the motion of human faces and handling the significant transitions of athletes. Notably, our approach excels at synthesizing high-quality videos within the challenging UCF101 dataset, a task where other comparative methods often falter. More visual results are available on the project page.

| Method | IS ↑ | FID ↓ |
|---|---|---|
| MoCoGAN | 10.09 | 23.97 |
| VideoGPT | 12.61 | 22.7 |
| MoCoGAN-HD | 23.39 | 7.12 |
| DIGAN | 23.16 | 19.1 |
| StyleGAN-V | 23.94 | 9.445 |
| PVDM | 60.55 | 29.76 |
| Latte (ours) | 68.53 | 5.02 |
| Latte+IMG (ours) | **73.31** | **3.87** |

Table 5: Inception Score and FID comparisons of Latte against other state-of-the-art on the UCF101 and FaceForensics datasets, respectively.

**Quantitative results.** In Tab. 3, we provide the quantitative results of Latte and other comparative methods, respectively. Our method significantly outperforms the previous works on all datasets, which shows the superiority of our method on video generation. In Tab. 5, we report the FID on FaceForensics and the IS on UCF101 to evaluate video frame quality. Here, "IMG" means video-image joint training. Our method demonstrates outstanding performance with an FID value of 3.87 and an IS value of 73.31, significantly surpassing the capabilities of other methods.

### 4.5 Extension to text-to-video generation

To explore the potential capability of our proposed method, we extend Latte to text-to-video generation. We adopt the method shown in Fig. 2 (a) to construct our Latte T2V model. Sec. 4.2 mentions that leveraging pre-trained models can facilitate model training. Consequently, we utilize the weights of pre-trained PixArt-$\alpha$ ($512 \times 512$ resolution) Chen et al. (2024a) to initialize the parameters of the spatial Transformer block in the Latte T2V model. Since the resolution of the commonly used video dataset WebVid-10M Bain et al. (2021) is lower than $512 \times 512$, we train our model on a high-resolution video dataset Vimeo25M proposed in Wang et al. (2024a). We compare with the recent T2V models, including VideoFusion Luo et al. (2023), VideoLDM Blattmann et al. (2023b), VideoCrafter He et al. (2023), CigVideo Hong et al. (2022), HiGen Qing et al. (2024), and OpenSoraPlan Lin et al. (2024) using the quantitative metrics established by VBench Huang et al. (2024), as shown in Table 4. It demonstrates that Latte can generate comparable T2V results.

## 5 Conclusion

This work presents Latte, a simple and general video diffusion method, which employs a video Transformer as the backbone to generate videos. To improve the generated video quality, we determine the best practices of the proposed models, including clip patch embedding, model variants, timestep-class information injection, temporal positional embedding, and learning strategies. Comprehensive experiments show that Latte achieves state-of-the-art results across four standard video generation benchmarks. In addition, text-to-video results are comparable to those of current T2V approaches. We strongly believe that Latte can

provide valuable insights for future research concerning the integration of transformer-based backbones into diffusion models for video generation as well as other modalities.

## Acknowledgments

This work is partially supported by the National Key R&D Program of China under Grant No. 2022ZD0160102, and the Science and Technology Commission of Shanghai Municipality under Grant No. 23QD1400800.

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

## A   Appendix

### A.1   The sampled video frames

We provide the sampled video frames of different methods as shown in Fig. 9 of Appendix.

### A.2   The structure of S-AdaLN

In Fig. 10 of Appendix, we show the structure of S-AdaLN.

### A.3 Discussion about the difference from concurrent works

A similar idea has been explored in recent concurrent work VDT Lu et al. (2024), GenTron Chen et al. (2024b), W.A.L.T Gupta et al. (2024), Open-Sora Plan Lin et al. (2024), HunyuanVideo Kong et al. (2024), Pyramidal Flow Jin et al. (2025), and so on. VDT, GenTron, and W.A.L.T use an architecture akin to our Variant 3. VDT primarily focuses on generating various video tasks, including image-to-video generation and unconditional video generation, utilizing a mask learning strategy. GenTron and W.A.L.T mainly focus on general purposes, i.e., text-to-video generation and text-to-image generation. Open-Sora Plan and HunyuanVideo focus on large-scale, open-source video generation models or system frameworks, respectively. Pyramidal Flow adopts the flow matching strategy, whose core idea and training paradigm are fundamentally different from diffusion-based generative methods. Its primary goal is to improve generation efficiency. Cosmos focuses on building a world foundation model for physical AI systems, such as robotics and autonomous driving. Its goal is to simulate the dynamic behaviors of the real physical world, emphasizing physical consistency. The primary difference from the previous works is that we conduct a systematic analysis of different Transformer backbones and the relative best practices discussed in Sec. 3.2 and Sec. 3.3 on video generation.

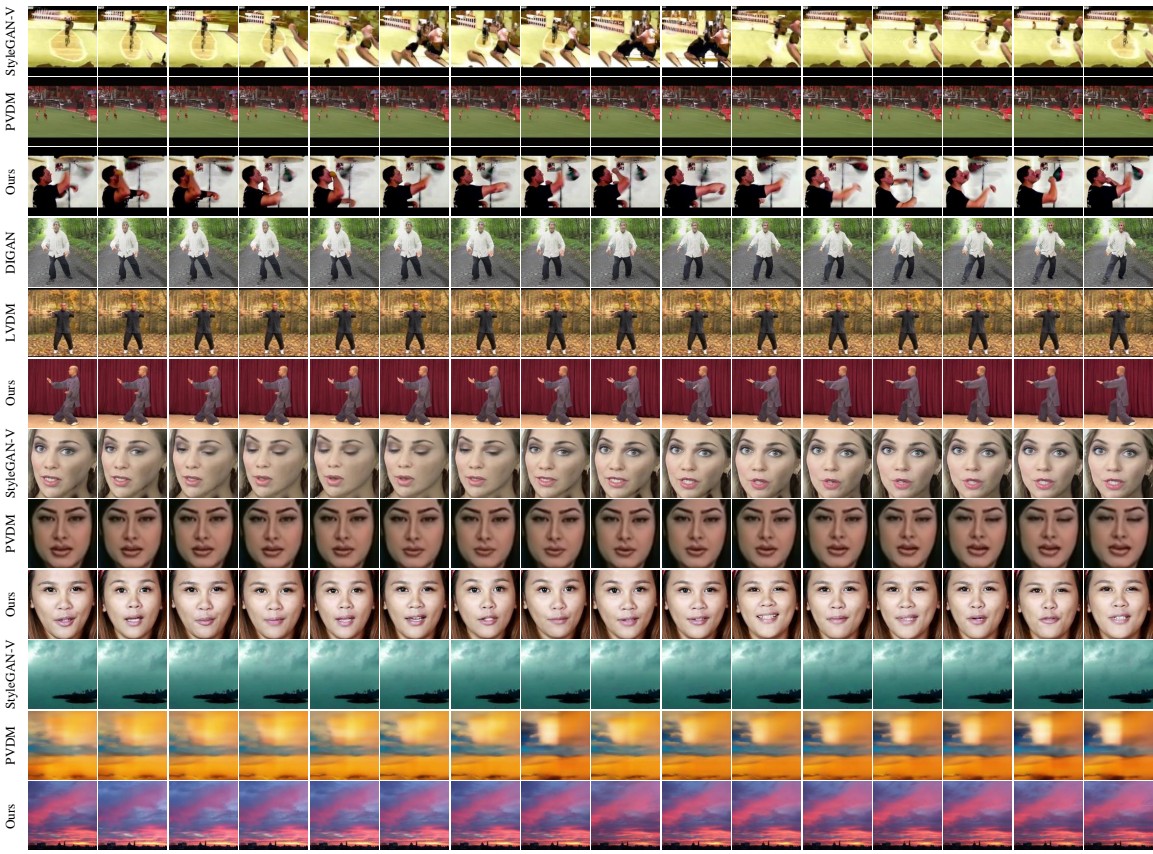

Figure 9: Sample videos from the different methods on UCF101, Taichi-HD, FaceForensics and SkyTimelapse, respectively.

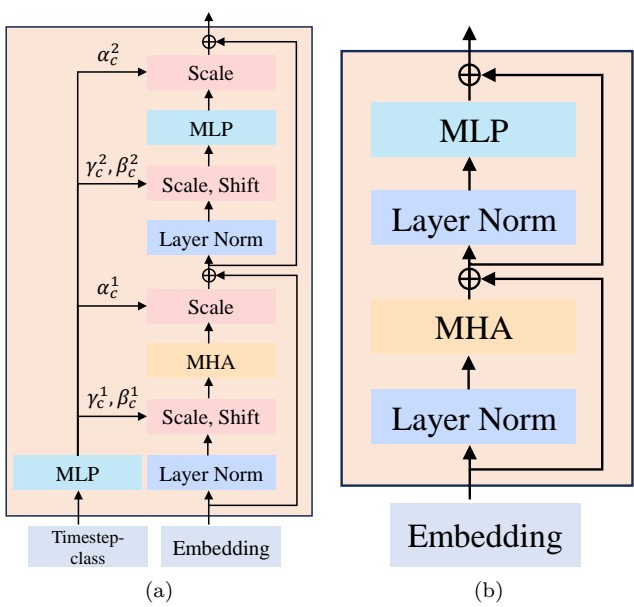

Figure 10: (a) The architecture of S-AdaLN described in Sec. 3.3.2. (b) The architecture of vanilla transformer block used in Fig. 2 (a) and (b). MLP and MHA mean the multi-layer perception layer and the multi-head attention, respectively.

