# OpenReview forum: "Latte: Latent Diffusion Transformer for Video Generation"
_TMLR — Accepted by TMLR_

### Review · Reviewer_LiRL · 2025-01-29

**Summary Of Contributions:**

The paper extends the transformer-based diffusion model (DiT), originally proposed for class-conditioned image generation, to (unconditional, class-conditional and text-conditioned) video generation.

The paper analyses several architectural designs and training approaches to handle the spatial-temporal tokens of video.

In summary, their exploration and findings are:

  	- Four variants of model architecture.
  	    - Variant 1 with alternating spatial and temporal attention performs best
  	- Two video patch embedding.
  		- Uniform frame patch performs better than compression frame patch embedding
  	- Timestep-class injection.
  		- Adaptive layer norm perfoms better than all-tokens
  	- Temporal positional embedding.
  		- absolute performs slightly better RoPE embedding
  	- Learning strategy.
  		- Randomly initialized model performs better than imagenet initialized
  		- Image-Video joint training performs better than video-only training

**Audience:**

Yes

**Broader Impact Concerns:**

Given the video generation model, the authors may discuss the possible misuse of the model.

**Claims And Evidence:**

Yes

**Requested Changes:**

Please refer to the weaknesses

**Strengths And Weaknesses:**

Strengths:
- The analysis of architecture comparison is new and essential to explore video generation using transformers.
- The generated videos in supplementary are realistic and good.

Major weaknesses:
- The rationale behind some design choices is not clearly explained and may be redundant.
  - Why do only these four model variants need to be compared? Especially, Variant 3 and 4.
  - In Sec 3.3.2, the need to compare "all-tokens" timestep-class injection should be better explained, maybe with some reference to previous work. Also, DiT (and several follow-up works [1,2]) have shown the benefit of Ada-LN.

- The conclusion about initializing with pre-trained weights is not fully explained. The authors say that due to differences in Imagenet and Faceforensics datasets, initializing with scratch is better. But what about other datasets?
  Contrary to this conclusion, their text-to-video model is intialised with Pix-art.

- Contrary to the outcome that absolute position embedding performs better than RoPE. A lot of recent video generation models (For eg. [3,4,5]) have been using RoPE instead of absolute encoding. More insights into preferring one over another should be experimented and discussed by the authors.

Missing related works:
A lot of recent works [3, 5, 6] around transformer model-based video generation have been proposed and the paper should mention it and compare with them.

Minor weakness:
- Table 6. Why is IS computed on UC101 and FID on Faceforensics?
- In 4.1, what does "UCF101, we use both training and testing splits" mean? Does this mean both splits are used for training? If yes, then FVD scores in Table 2 would be unfair.

- I guess, Page 5, Variant 1, zt ∈ Rt×nf ×d should be zt ∈ Rs×nf ×d
- The paper should rearrange the tables and figures to better follow the text.
	- Table 5 is referred before Table 4.
  	- In Sec. 3.3.2, Fig. 10 is referred before Fig. 4
  	- Sequence of figures in Figure 5, For eg. Fig 5e is referred before 5a,b,c,d

[1] Ma et al, "SiT: Exploring Flow and Diffusion-based Generative Models with Scalable Interpolant Transformers", ECCV 2024

[2] Chen et al, "GenTron: Diffusion Transformers for Image and Video Generation", CVPR 2024

[3] Jin et al, "PYRAMIDAL FLOW MATCHING FOR EFFICIENT VIDEO GENERATIVE MODELING", ArXiv, 2024

[4] Yang et. al "CogVideoX: Text-to-Video Diffusion Models with An Expert Transformer", arXiv 2024

[5] Open-Sora Plan Team "Open-Sora Plan: Open-Source Large Video Generation Model", arXiv, 2024

[6] "HunyuanVideo: A Systematic Framework For Large Video Generative Models"

---

> ### Author Response · Authors · 2025-02-10
> **Respond to Reviewer LiRL 1/2**
>
> We would like to thank Reviewer LiRL for the valuable feedback.
>
> **Q1.** The rationale behind some design choices is not clearly explained and may be redundant.
>
> **Q1-1.** Why do only these four model variants need to be compared? Especially, Variant 3 and 4.
>
> **A1-1.** In Sec. 4.2 of the revised manuscript, we have thoroughly discussed the motivation for designing four variants. We would like to answer a key question: to what extent should the temporal and spatial modules be decoupled in a video diffusion transformer?
>
> *Excessive consecutive spatial attention modules can impair temporal coherence.* In Fig.6 of the revised manuscript, we analyze Variant 2 by computing the cosine similarity matrix between frame features at each block. The results show that adding spatial attention modules decreases the mean inter-frame cosine similarity while increasing its variance. This uneven decline leads to inverted relationships, where, for example, the similarity between the first and second frames is lower than that between the first and fifth. These inversions indicate disrupted temporal coherence, likely caused by ineffective decoupling of spatiotemporal information.
>
> *The features of the output frames from spatial attention do not match those from temporal attention.* Our analysis challenges the design of stacking numerous spatial attention modules, prompting us to explore tighter coupling between spatial and temporal modules. Thus, we propose Variant 4, which replaces the alternating spatiotemporal structure with a parallel one. However, experiments show poor performance due to the direct fusion of mismatched frame features. As illustrated in Fig. 8 in the revised manuscript, inter-frame cosine similarity distributions for temporal and spatial attention differ significantly, with extremely low similarity between them. Directly adding these features severely disrupts temporal information.
>
> The design of Variant 3 is derived from TimesFormer. Unlike Variant 1, which also employs an alternating spatiotemporal attention structure, Variant 3 has a higher degree of coupling. It directly feeds spatial attention features into temporal attention with only layer normalization. However, our analysis suggests this approach is suboptimal. The simple normalization of spatial attention frame features is insufficient to make them suitable for the temporal attention layer. Experiments confirm that while Variant 3 outperforms both Variant 2 and 4, its performance is still not as good as Variant 1, reinforcing our findings.
>
> **Q1-2.** In Sec 3.3.2, the need to compare "all-tokens" timestep-class injection should be better explained, maybe with some reference to previous work. Also, DiT (and several follow-up works [1,2]) have shown the benefit of Ada-LN.
>
> **A1-2.** The trend in multimodal large models is to concatenate the tokens from different modalities at the input stage [7, 8]. Inspired by this technique, we explore a similar approach in our paper, where we concatenate all conditional information with the video tokens at the input stage. While works like DiT and SiT have demonstrated the advantages of AdaLN in video and image generation, they do not compare it with the all-tokens approach. In contrast, we systematically evaluate the performance differences at the fusion stage of conditional information (input-stage concatenation vs. intermediate-layer injection), addressing the limitations of existing research that solely focuses on optimizing normalization layers.

---

> ### Author Response · Authors · 2025-02-10
> **Respond to Reviewer LiRL 2/2**
>
> **Q2.** The conclusion about initializing with pre-trained weights is not fully explained. The authors say that due to differences in Imagenet and Faceforensics datasets, initializing with scratch is better. But what about other datasets? Contrary to this conclusion, their text-to-video model is intialised with Pix-art.
>
> **A2.** Our conclusion regarding model initialization strategies is indeed context-dependent and requires further clarification. The recommendation for training from scratch specifically applies to scenarios involving small-scale datasets (e.g., UCF101, FaceForensics, SkyTimelapse, and Taichi-HD) where significant domain gaps exist between ImageNet and target video domains. In such cases, we empirically observe that the domain discrepancy may outweigh the benefits of knowledge transfer.
>
> However, for large-scale video generation tasks (e.g., text-to-video generation) where the training data encompasses broader visual domains, we adopt pre-trained foundation models (Pix-Art in this case) based on two key considerations:
>
> - The diversity of large-scale datasets inherently reduces domain gaps compared to specialized small datasets;
> - Pre-trained weights provide substantial advantages in training efficiency, accelerating convergence while maintaining comparable visual performance.
>
> This distinction aligns with our discussion in Section 4.2 regarding the trade-off between domain adaptation cost and knowledge transfer benefits. We have clarified this contextual dependence more explicitly in the revised manuscript to avoid potential misunderstanding.
>
> **Q3.** Contrary to the outcome that absolute position embedding performs better than RoPE. A lot of recent video generation models (For eg. [3,4,5]) have been using RoPE instead of absolute encoding. More insights into preferring one over another should be experimented and discussed by the authors.
>
> **A3.** In our study, the choice of absolute position encoding over RoPE was empirically driven by quantitative evaluations—our experiments showed that absolute position encoding achieved slightly better performance in terms of FVD on the FaceForensics dataset.
>
> While RoPE has been widely adopted in recent video generation works, these approaches primarily focus on dynamic temporal-spatial adaptability to generate variable-duration and multi-resolution videos. In contrast, our framework is designed for fixed-resolution video synthesis with predetermined temporal lengths, where the architectural advantages of RoPE’s relative positioning mechanism are less critical. This fundamental difference in design objectives may explain the performance discrepancy observed in our implementation.
> We recognize this distinction as a key methodological consideration and have expanded on this discussion in the revised manuscript.
>
> **Q4.** Missing related works: A lot of recent works [3, 5, 6] around transformer model-based video generation have been proposed and the paper should mention it and compare with them.
>
> **A4.** We have mentioned references [3, 5, 6] in the Related Work section and compared Open-Sora Plan with Latte in Tab. 4 in the revised manuscript.
>
> **Q5.** Table 6. Why is IS computed on UC101 and FID on Faceforensics?
>
> **A5.** We follow the evaluation settings of StyleGAN-V and PVDM, choosing to compare IS and FID metrics against other methods on UCF101 and FaceForensics.
>
> **Q6.** In 4.1, what does "UCF101, we use both training and testing splits" mean? Does this mean both splits are used for training? If yes, then FVD scores in Table 2 would be unfair.
>
> **A6.** For UCF101, we train our model using both the training and test splits. However, the comparison remains fair, as we adopt the same approach as previous methods, such as StyleGAN-V, and utilize the identical splits for training.
>
> **Q7.** I guess, Page 5, Variant 1, zt ∈ Rt×nf ×d should be zt ∈ Rs×nf ×d
>
> **A7.** We thank the reviewer for pointing it out. We have revised it.
>
> **Q8.** The paper should rearrange the tables and figures to better follow the text.
>
> **A8.** We have reorganized the figures to align with the text as much as possible.
>
> **References:**
>
> [1] Ma et al, "SiT: Exploring Flow and Diffusion-based Generative Models with Scalable Interpolant Transformers", ECCV 2024
>
> [2] Chen et al, "GenTron: Diffusion Transformers for Image and Video Generation", CVPR 2024
>
> [3] Jin et al, "PYRAMIDAL FLOW MATCHING FOR EFFICIENT VIDEO GENERATIVE MODELING", ArXiv, 2024
>
> [4] Yang et. al "CogVideoX: Text-to-Video Diffusion Models with An Expert Transformer", arXiv 2024
>
> [5] Open-Sora Plan Team "Open-Sora Plan: Open-Source Large Video Generation Model", arXiv, 2024
>
> [6] "HunyuanVideo: A Systematic Framework For Large Video Generative Models"
>
> [7] "Boosting Multimodal Large Language Models with Visual Tokens Withdrawal for Rapid Inference", AAAI 2025
>
> [8] "Visual Instruction Tuning", NeurlPS 2023

---

### Review · Reviewer_Ctig · 2025-02-02

**Summary Of Contributions:**

This paper conducts detailed experimental analysis about the transformer-based video diffusion model. To incorporate spatial and temporal information of video generation, the author proposes four variants of self-attention layer. The proposed model demonstrates better generation performance than existing video generation methods.

**Audience:**

Yes

**Broader Impact Concerns:**

Though broader impact is not discussed, since the paper is about experimental evaluation of the model architecture and training method, I think the ethical risk of the paper is small.

**Claims And Evidence:**

Yes

**Requested Changes:**

Discussion and Comparison to existing transformer-based video diffusion methods.

Summarization of the experimental findings.

**Strengths And Weaknesses:**

Strength

The author proposes four variants of self attention layers (i) alternately applying spatial and temporal self attention layers (ii) first applies spatial attention blocks and then applies temporal attention blocks (iii) alternatively applying spatial and temporal multi head attention blocks (iv) decomposition of the multi head attention blocks into spatial heads and temporal heads. The author experimentally evaluates and discusses that the variant (i) is the best.

Further, the author proposes several best practices for the generative model including temporal positional embedding and image-video joint training.

The author conducts several experimental evaluations including ablation study. The proposed method demonstrates better generation performance than the existing generation methods on several benchmark datasets.


Weakness

As the author says, the transformer-based diffusion video generation has already been proposed as VDT and GenTron. It would be better to add more detailed discussion about the difference from these works and experimental comparison.

Since the author conducts various experimental evaluation, it may be clearer to add the section that summarizes the experimental findings e.g. in conclusion or before conclusion section.

---

> ### Author Response · Authors · 2025-02-09
>
> We would like to thank Reviewer Ctig for the valuable feedback.
>
> **Q1.**  As the author says, the transformer-based diffusion video generation has already been proposed as VDT and GenTron. It would be better to add more detailed discussion about the difference from these works and experimental comparison.
>
> **A1.** We have added a comparison in Appendix Sec. A.3, highlighting the differences between Latte and several Transformer-based video generation methods, including VDT, GenTron, W.A.L.T [1], Open-Sora Plan [2], Pyramidal Flow [3], and Cosmos [4]. The comparison covers key aspects such as methodology, objectives, and experimental settings. Additionally, we have included a quantitative comparison with W.A.L.T and OpenSoraPlan in Tables 3 and 4 of the revised manuscript.
>
> **Q2.** Since the author conducts various experimental evaluation, it may be clearer to add the section that summarizes the experimental findings e.g. in conclusion or before conclusion section.
>
> **A2.** We have added a summary of best practices in Sec. 4.3 of the revised manuscript.
>
> **References:**
>
> [1] Gupta, Agrim, et al. "Photorealistic video generation with diffusion models." European Conference on Computer Vision. Cham: Springer Nature Switzerland, 2024.
>
> [2] Lin, Bin, et al. "Open-sora plan: Open-source large video generation model." arXiv preprint arXiv:2412.00131 (2024).
>
> [3] Jin, Yang, et al. "Pyramidal flow matching for efficient video generative modeling." arXiv preprint arXiv:2410.05954 (2024).
>
> [4] Agarwal, Niket, et al. "Cosmos world foundation model platform for physical ai." arXiv preprint arXiv:2501.03575 (2025).

---

### Review · Reviewer_CAc1 · 2025-02-04

**Summary Of Contributions:**

This paper explores the design choices for spatio-temporal transformer blocks in latent video diffusion models. Specifically, it reveals that the interleaved fusion design, which alternates spatial and temporal attention blocks, performs the best among other designs. In addition, the paper ablates the choices for patch embedding, positional embedding, timestep injection, and video-image joint training. Combining all together, the proposed Latte achieves SOTA for unconditional and class-conditional video generation benchmarks, while also being effective for text-to-video generation.

**Audience:**

Yes

**Claims And Evidence:**

Yes

**Requested Changes:**

I think the paper is solid and carefully conducted, making it a strong contribution to TMLR. My only suggestion is to add the latest works in the benchmark tables to ensure a more comprehensive comparison.

**Strengths And Weaknesses:**

### Strengths

1. Careful ablation study over design choices – In particular, I can see the effort of the authors in claiming that it is more than just VLDM + Timesformer.
2. Find the proper spatio-temporal transformer design – Interleaved fusion intuitively performs the best.
3. Strong empirical results – Benchmarking is conducted on various setups, showing consistent improvements.

---
### Weaknesses

1. Limited technical novelty – The main ideas of most components are from prior work, e.g., AdaLN from DiT, though the paper does a great job at engineering to carve out performance.
2. Benchmark tables could be more up-to-date – The paper mainly compares against existing models but does not include some of the latest works. For example, WARP (Gupta et al., ECCV 2024) reports an FVD of 258.1 on UCF-101, and Latte (202.23) is actually better. Yet, adding more recent models would help give a more extensive and proper sense of comparisons.

---

> ### Author Response · Authors · 2025-02-09
>
> We would like to thank Reviewer CAc1 for the valuable feedback.
>
> **Q1.** Limited technical novelty – The main ideas of most components are from prior work, e.g., AdaLN from DiT, though the paper does a great job at engineering to carve out performance.
>
> **A1.** Our main contributions are reflected in the systematic exploration of how to effectively design architectures and learning strategies of Transformer-based video diffusion models. Additionally, in Sec. 4.2 of the revised manuscript, we provide a deeper analysis of the design principles of the four model variants from the perspective of how temporal and spatial modules decouple in video generation models. Moreover, video generation models in the open-source community, including but not limited to OpenSora and OpenSora Plan, have extensively drawn inspiration from the design concepts of Latte, as Latte offers significant technical insight and novelty.
>
> **Q2.** Benchmark tables could be more up-to-date – The paper mainly compares against existing models but does not include some of the latest works. For example, WARP (Gupta et al., ECCV 2024) reports an FVD of 258.1 on UCF-101, and Latte (202.23) is actually better. Yet, adding more recent models would help give a more extensive and proper sense of comparisons.
>
> **A2.** We believe the reviewer would like to mention W.A.L.T. [1] rather than WARP. We have updated the comparisons on UCF101 to include W.A.L.T as shown in Tab. 3 in the revised manuscript.
>
> **References:**
>
> [1] Gupta, Agrim, et al. "Photorealistic video generation with diffusion models." European Conference on Computer Vision. Cham: Springer Nature Switzerland, 2024.

---

### Author Response · Authors · 2025-03-17
**Inquiry Regarding Discussion and Final Recommendation for Latte**

We sincerely appreciate the reviewers and the Action Editor for their time and effort in evaluating our paper. We would like to inquire whether our submitted rebuttal has adequately addressed the concerns raised by the reviewers. If there are any remaining issues, we would be more than happy to engage in further discussion. If our responses have sufficiently addressed the concerns of the reviewers, we would sincerely appreciate your consideration for a positive final recommendation when possible.

Best Regards,

Latte Team

---

### Decision · Action_Editor_UHsE · 2025-03-16

**Recommendation:** Accept with minor revision

**Comment:**

All reviewers consider the novelty of this work to be limited, but the majority appreciate the experimental evaluations/findings and proposed techniques are useful to the community. The editor agrees with most reviewers and recommends it be accepted with the following revisions:
* The authors should revise this paper based on the comments and responses.
* The authors should revise the claims and emphasize empirical contributions rather than algorithmic ones.
* The authors are strongly encouraged to make the source codes and trained models available to the public as those are the main contributions of this work, and would make this work more impactful.

**Audience:**

This paper is of broad interest to vision and learning researchers interested in video generation, transformers, and representation learning from videos.

**Claims And Evidence:**

The main claim of this paper is a novel Latent Diffusion Transformer for video generation that achieves state-of-the-art performance. The Latte model extracts spatio-temporal tokens from input videos and then adopts a series of Transformer blocks to model video distribution in the latent space. Numerous modifications have been added to improve model performance, including clip patch embedding, model variants, timestep-class information injection, temporal position embedding, and learning strategies.